# fMRI reveals neural activity overlap between adult and infant pain

Sezgi Goksan[1], Caroline Hartley[2], Faith Emery[2], Naomi Cockrill[2], Ravi Poorun[1], Fiona Moultrie[2], Richard Rogers[1], Jon Campbell[1], Michael Sanders[1], Eleri Adams[2], Stuart Clare[1], Mark Jenkinson[1], Irene Tracey[1], Rebeccah Slater[1,2]*

[1]Oxford Centre for Functional Magnetic Resonance Imaging of the Brain, Nuffield Department of Clinical Neurosciences, University of Oxford, Oxford, United Kingdom; [2]Department of Paediatrics, University of Oxford, Oxford, United Kingdom

**Abstract** Limited understanding of infant pain has led to its lack of recognition in clinical practice. While the network of brain regions that encode the affective and sensory aspects of adult pain are well described, the brain structures involved in infant nociceptive processing are less well known, meaning little can be inferred about the nature of the infant pain experience. Using fMRI we identified the network of brain regions that are active following acute noxious stimulation in newborn infants, and compared the activity to that observed in adults. Significant infant brain activity was observed in 18 of the 20 active adult brain regions but not in the infant amygdala or orbitofrontal cortex. Brain regions that encode sensory and affective components of pain are active in infants, suggesting that the infant pain experience closely resembles that seen in adults. This highlights the importance of developing effective pain management strategies in this vulnerable population.

*For correspondence: rebeccah.
slater@paediatrics.ox.ac.uk

**Competing interests:** The
authors declare that no
competing interests exist.

**Reviewing editor:** Jody C
Culham, University of Western
Ontario, Canada

## Introduction

The network of brain regions that encode both the affective and sensory aspects of the pain experience have been well described in the adult (*Apkarian et al., 2005*; *Tracey and Mantyh, 2007*). It is not known which cortical and subcortical brain structures are activated following noxious events in infants. Early evidence demonstrated that infants exhibited reflex responses and concluded that pain was not processed at the level of the cortex (*Rodkey and Pillai Riddell, 2013*). This, coupled with an infant's inability to describe their pain experience verbally, led to extreme controversy regarding whether an infant has the ability to experience the unpleasant affective components of pain (*Rodkey and Pillai Riddell, 2013*). Consequently, infants have received poor pain management, exemplified during the 1980s by surgery being routinely performed using neuromuscular blocks without provision of adequate analgesia (*Anand and Hickey, 1987*). More recent research has primarily focussed on behavioural and physiological measures, which has led to the development of a number of infant pain assessment tools (*Duhn and Medves, 2004*). However, the lack of sensitivity and specificity of these measures means the trend to undertreat pain remains in clinical practice (*Carbajal et al., 2008*), despite concerted efforts to improve the management of pain in this population (*Anand and International Evidence-Based Group for Neonatal Pain, 2001*). For example, it is remarkable that current UK NHS guidelines for ankyloglossia (tongue tie) surgery state that 'in small babies, being cuddled and fed are more important than painkillers' (*NHS Choices, 2015*). Indeed, a recent review of neonatal pain management practice in intensive care highlighted that although infants experience an average of 11 painful procedures per day, 60% of the population did not receive any pharmacological analgesia (*Roofthooft et al., 2014*).

Recent studies using EEG and near-infrared spectroscopy have been used to provide reliable evidence that nociceptive information is transmitted to the newborn infant brain

**eLife digest** Doctors long believed that infants do not feel pain the way that older children and adults do. Instead, they believed that the infants' responses to discomfort were reflexes. Based on these beliefs, it was a routine practice to perform surgery on infants without suitable pain relief up until the late 1980s. Even now, infants may receive less than ideal pain relief. For example, a review found that although newborns in intensive care units undergo 11 painful procedures per day on average, more than half of the babies received no pain medications. Some guidelines continue to emphasize that for infants cuddling and feeding are more important sources of comfort than pain-relieving drugs.

There is growing support for better pain control for infants. Doctors and nurses now routinely observe behaviour and physiological responses—such as heart rate—to assess whether infants are experiencing pain. When an infant shows signs of pain, medical staff may give the infant sugar water or other interventions aimed at reducing their distress. However, recordings of brain activity suggest that infants may experience pain without exhibiting physical signs and that sugar water may reduce the behaviours associated with pain but not the pain itself.

More objective measurements of infant pain would be useful, but to create such measurements scientists must first understand how infants experience pain. So Goksan et al. used a technique called functional magnetic resonance imaging (fMRI) to compare the brain responses of adults and newborns to the same stimulus—a sharp poke of the foot. The adults were also asked about the pain they experienced, and whether the infants pulled their foot away when poked was documented.

The fMRI results revealed that pain increased activity in 20 regions in the adults' brains, and 18 of the same regions in the infants' brains. The brain regions activated in the infants' brains in response to a poke on the foot are involved in processing sensations and emotions. The two regions that did not activate in the infant brains—the amygdala and the orbitofrontal cortex—help individuals interpret the stimuli. Goksan et al. therefore conclude that infants experience pain in similar ways to adults, though they may not experience all the emotions that adults have when they are in pain. It is, therefore, important to give infants suitable pain relief during potentially painful procedures.

(*Slater et al., 2006*, *2010a*, *2010b*), and have highlighted the limitations of using observational behavioural measures to quantify pain in infants. For example, nociceptive information can be processed in the infant brain without a concomitant behavioural response (*Slater et al., 2008*), and interventions thought to alleviate pain (i.e., oral sucrose) can reduce clinical pain scores without reducing evoked nociceptive brain activity (*Slater et al., 2010a*). While these studies confirm that the infant central nervous system can process noxious stimulation, they do not elucidate the nature of the infant experience—in particular, which brain regions are involved, and therefore, whether the sensory, cognitive, and emotional aspects of pain are present in this population. Here, we identify the cortical and subcortical structures activated following acute noxious stimulation in the healthy newborn infant, and compare the activity with that observed in adults. The feasibility of this approach was demonstrated in a foundational pilot study (*Williams et al., 2015*). A case study in a single infant demonstrated that noxious stimulation evoked widespread brain activity (*Williams et al. 2015*), which included brain regions previously reported to be involved in adult pain (*Tracey et al., 2007*). Using a reverse inference approach to compare active brain regions in infants with those reported during adult pain, we postulate which aspects of the pain experience are present (*Wager et al., 2013*), providing an opportunity to gain insight into the organisation of nociceptive circuitry in the naïve infant brain.

In this study, acute noxious stimulation (PinPrick Stimulators, MRC Systems) was applied to the foot in both adults (n = 10; applied force: 32–512 mN) and infants (n = 10; applied force: 32–128 mN; greater force was not applied due to the potential risk of tissue damage). Using functional magnetic resonance imaging (fMRI) changes in blood oxygen level dependent (BOLD) activity in the brain were recorded in response to the stimuli. Adults were asked to verbally report their pain intensity and, using the McGill Pain questionnaire (*Melzack and Torgerson, 1971*), to describe the quality of the

pain they experienced. As infants are unable to describe their pain, reflex leg withdrawal from the stimuli was visually observed during scanning. Parents were present during the studies and no infants were withdrawn from the study after recruitment.

## Results and discussion

Adult participants reported increased pain with increasing stimulus intensity (r = 0.48; p < 0.0001), and most frequently described the pain as pricking (n = 8 of 10) and sharp (n = 6 of 10). In infants, application of the stimuli evoked visible withdrawal of the stimulated leg, which could be observed at all stimulus intensities, whereas in adults, reflex withdrawal of the leg or foot was not observed at any stimulus intensity. While low threshold stimuli can also evoke reflex withdrawal in infants (*Cornelissen et al., 2013*), this observation confirms that the stimuli applied in this study were detected by the peripheral nervous system and transmitted to the central nervous system. Although noxious stimulation can elicit reflex limb withdrawal in adults, supraspinal modulation of the input means this activity is often suppressed in experimental studies.

In adults, noxious stimulation evoked significant increases in BOLD activity in cortical and subcortical brain regions, including primary somatosensory cortices, anterior cingulate cortex (ACC), bilateral thalamus, and all divisions of the insular cortices (*Figure 1*). All brain regions that had a significant increase in BOLD following noxious stimulation are identified in *Table 1*, and are consistent with previous literature (*Tracey and Mantyh, 2007*). In infants, the increases in BOLD activity evoked by the noxious stimulation were extremely similar to that seen in adults, and all but two of the 20 regions that were active in the adults were active in infants (*Table 1* and *Figure 1*). While in adults the parietal lobe, pallidum, and precuneus cortex were only active in the brain regions contralateral to the site of stimulation, in infants these brain regions were also active on the ipsilateral side to the stimulus. Additional brain regions that were only active in the infants included the bilateral auditory cortices, hippocampus, and caudate (*Table 1*). The increased bilateral activity and greater number of active regions in infants are likely due to the immature cortico-cortical and interhemispheric

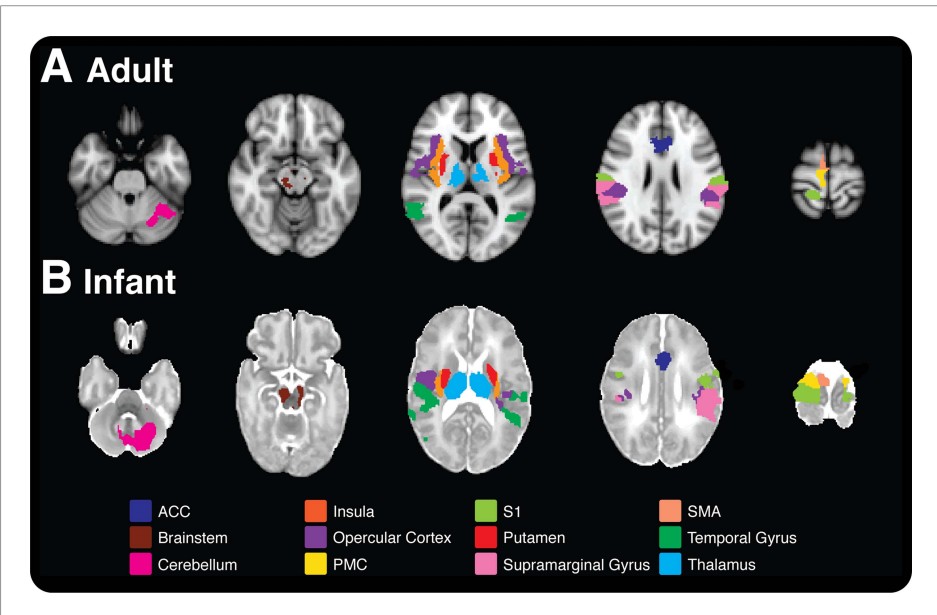

**Figure 1**. Comparison of nociceptive-evoked brain activity in selected brain regions that are active in both adults and infants. Significantly, active voxels across each stimulus intensity level are presented for (**A**) adult and (**B**) infant participants (applied force: adults 32–512 mN; infants 32–128 mN). Each colour represents activity in a different anatomical brain region. (**A**) Adult activity is overlaid onto a standard T1 weighted MNI template and (**B**) infant activity is overlaid onto a standard T2 weighted neonatal template, corresponding to a 40-week gestation infant. ACC: anterior cingulate cortex; S1: primary somatosensory cortex: PMC: primary motor cortex; SMA: supplementary motor area.

pathways (*Kostovic and Jovanov-Milosevic, 2006*). Major reorganisation of the cortical circuitry occurs after the first postnatal month when there is a striking retraction of exuberant axons in the corpus callosum and there is a cessation of growth of the long cortico-cortical afferent pathways (*Jovanov-Milosevic et al., 2006*; *Kostovic and Jovanov-Milosevic, 2006*).

Although the infant brain activity was widespread, the specificity of the response was demonstrated, as it was not present across all brain regions. For example, brain regions not commonly associated with the cerebral processing of nociceptive stimulation in the adult, such as the olfactory cortex, cuneus, and fusiform gyrus, were also not active in the infants. 14% of voxels across the whole brain were active following the application of the 128 mN stimuli in infants compared with 9% of voxels following the 512 mN stimuli in adults (*Figure 2*). In contrast, the 128 mN stimulus activated less than 1% of voxels in the adult brain. This demonstrates that the coverage and distribution of brain activity evoked by the 128 mN stimulus in infants was most similar to that evoked by the 512 mN stimulus in adults (*Figure 2*). This suggests that infants have increased sensitivity to nociceptive stimuli compared with adults, which is supported by previous data that show spinal nociceptive reflex withdrawal activity has greater amplitude and duration in infants compared with adults (*Andrews and Fitzgerald, 1999*; *Skljarevski and Ramadan, 2002*; *Cornelissen et al., 2013*). These data strongly imply that the threshold for evoking widespread nociceptive brain activity in infants is substantially lower than in adults. It is, however, not known whether the increased brain activity observed at a lower threshold in the infants is due to increased peripheral drive, for example due to differences in skin thickness between the adult and infant populations, or due to differences in transduction or subsequent central processing of the nociceptive input.

Noxious stimulation in infants did not evoke activity in the amygdala or orbitofrontal cortex (OFC) (*Table 1*), and in contrast to the adults, where activity was present across all divisions of bilateral insular cortices, activity in the anterior division was not present (*Figure 1*). A recent white matter tractography study of the adult brain shows that the anterior insula has dominant connections with the OFC (*Wiech et al., 2014*). Based on many imaging studies spanning a range of stimuli and tasks, it is thought that activation in the anterior insula reflects the net evaluation of the affective impact of an impending situation. Similarly, the OFC is sensitive to stimuli with an emotional valence, however, it primarily responds to the reward value of the stimulus (including negative value) rather than its sensory features. Importantly, the OFC also encodes the anticipation of future outcomes, which makes it well suited for guiding subsequent decisions (*Kahnt et al., 2010*). It is likely that the infants are too immature and inexperienced to evaluate and contextualise the nociceptive stimulus into a coordinated decision and response, which might account for the lack of activity within these regions. Similarly, in adults the amygdala is thought to attach emotional significance to the nociceptive inputs it receives, and to play a role in fear and anxiety (*Simons et al., 2014*), which may reflect affective qualities that the newborn infant does not yet ascribe to the stimulus.

In light of these observations, it is plausible that infants do not experience the full range of aversive qualities that adults associate with nociceptive input. Indeed, this hypothesis is supported by evidence from rat pups, which shows that avoidance behaviour in a fear-conditioning paradigm does not manifest until postnatal day 10, and is associated with the enhancement of neural activity within the amygdala (*Sullivan et al., 2000*; *Sullivan, 2001*). Nevertheless, the observation that brain structures involved in affective processing, such as the anterior cingulate cortex, are activated following noxious stimulation suggests that infants do have the capacity to experience an emotionally relevant context related to incoming sensory input. Indeed, in adults the modulation of pain-related activity in the anterior cingulate cortex closely parallels a selective change in perceived unpleasantness (*Rainville et al., 1997*).

11 brain regions significantly encoded stimulus intensity in adults, whereas none of the active regions in infants exhibited significant intensity encoding (*Table 1*). Although the trend for intensity encoding in infants is clearly evident in some brain regions, these data suggest that infants do not discriminate stimulus intensity as well as adults (*Figure 2—figure supplement 1*). As only three stimulus intensities were applied to the infants it is plausible that if the intensity range were increased, significant intensity encoding may be observed. Nevertheless, when considering adult brain regions that did significantly intensity encode, three of the four highest ranked brain regions (ranked according to the degree of intensity encoding, and identified as the contralateral temporal gyri, opercular cortex, and all divisions of the insular cortex), were ranked in the same order within the top three regions in infants, highlighting the remarkable similarity in how the immature infant brain and

adult brain encode nociceptive information (*Table 1*). Intensity encoding has been reported following low intensity von Frey hair stimulation (*Williams et al., 2015*).

Inferences about the subjective experience of pain are highly speculative, whether based on brain imaging data, behavioural responses or other autonomic or physiological observations. In most adults, where the pain experience can be communicated verbally, it is not always necessary to rely on surrogate measures when attempting to quantify an individual's pain experience or when assessing the need for analgesic provision. However, where verbal report is not possible as in the infant population or in those who are cognitively impaired, reliance on surrogate measures is essential when making inferences about pain perception. As cortical activation is a fundamental requirement for an experience to be interpreted as painful, inferences based on patterns of brain activity may provide the most reliable surrogate measure of pain compared with alternative approaches based on behavioural and physiological indicators that may not be reliably linked to central sensory or emotional processing in the brain (*Oberlander et al., 2002*; *Ranger et al., 2007*). This does not, however, negate the importance of taking a multidimensional approach to infant pain assessment by considering measures of brain activity in the context of other well-characterised behavioural and physiological indicators. Indeed, some researchers have argued that reverse inference based on brain imaging results should be used merely as a guide to direct further enquiry rather than a direct means to interpret results (*Poldrack, 2008*). Nevertheless, it has been shown using multivariate pattern analysis that pain-related brain activity can be classified and discriminated from other psychological states, suggesting a neural state for pain perception that is distinct from other sensory modalities and affective experiences (*Yarkoni et al., 2011*; *Wager et al., 2013*). Although we cannot necessarily infer an infant's subjective experience based on a given pattern of brain activity, these results make certain conclusions more likely. The closer the pattern of brain activity mimics activity observed in adults—who can report their subjective experience—the stronger the

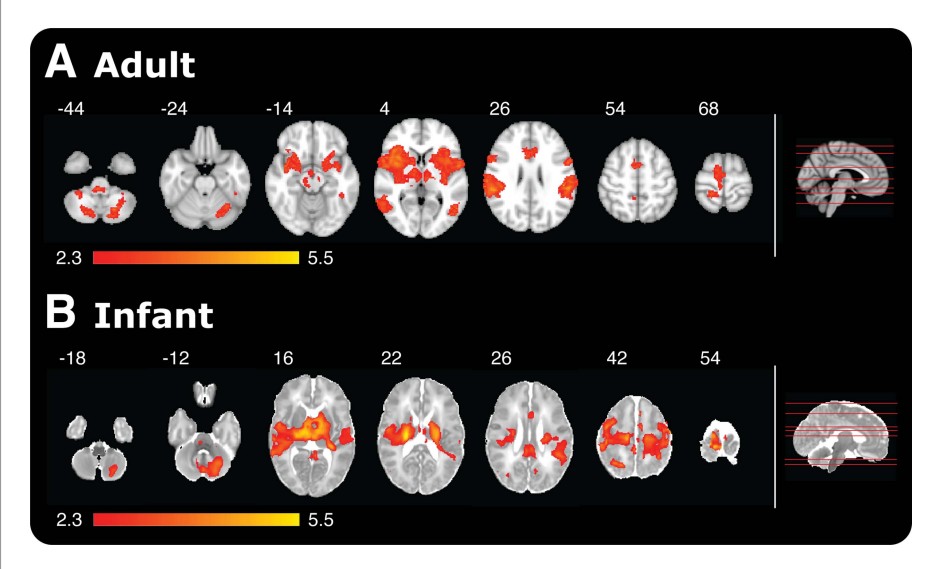

**Figure 2**. Noxious-evoked brain activity in response to the maximal presented stimulus in adults (512 mN) and infants (128 mN). Red-yellow coloured areas represent active brain regions (threshold $z \geq 2.3$ with a corrected cluster significance level of $p < 0.05$). An image of a midline sagittal brain slice (right panel) identifies the location of each example slice in the horizontal plane. (**A**) Adult activity is overlaid onto a standard T1 weighted MNI template and (**B**) infant activity is overlaid onto a standard T2 weighted neonatal template, corresponding to a 40-week gestation infant.

The following figure supplement is available for figure 2:

**Figure supplement 1**. Relationship between percentage change in BOLD signal and stimulus intensity (force) in four example active brain regions in adult and infant participants.

**Table 1.** Identification of all active brain regions in adults and infants following acute noxious stimulation at all stimulus intensities (applied force: adults 32–512 mN; infants 32–128 mN)

| | Anatomical area | Region | Adults Peak Z within cluster | MNI coords x | y | z | Rank | Slope of regression (*E-03) | P val* | Infants Peak Z within cluster | Neonate template coords x | y | z | Rank | Slope of regression (*E-03) | P val* |
|---|---|---|---|---|---|---|---|---|---|---|---|---|---|---|---|---|
| Intensity encoding regions (in adults) | Temporal gyrus | Contra | 3.92 | 64 | −34 | 20 | 1 | 1.01 | 0.0002 | 3.05 | 32 | −32 | 12 | 1 | 2.46 | 0.0083 |
| | Cingulate gyrus | Anterior | 4.11 | 6 | 4 | 40 | 2 | 0.65 | 0.0005 | 2.58 | −1 | 1 | 26 | 11 | 1.01 | 0.3971 |
| | Opercular cortex | Contra | 5.60 | 40 | 6 | 10 | 3 | 0.63 | 0.0001 | 3.38 | 32 | −13 | 19 | 2 | 2.23 | 0.0391 |
| | Insula | Contra | 4.18 | 34 | 14 | 6 | 4 | 0.61 | 0.0001 | 3.04 | 19 | −22 | 23 | 3 | 2.15 | 0.0207 |
| | Supramarginal gyrus | Contra | 4.33 | 64 | −38 | 20 | 5 | 0.60 | 0.0008 | 3.29 | 25 | −23 | 39 | 9 | 1.08 | 0.1749 |
| | Postcentral gyrus | Contra | 4.28 | 58 | −18 | 22 | 6 | 0.60 | 0.0012 | 3.85 | 15 | −22 | 52 | 10 | 1.01 | 0.2667 |
| | Visual cortex | Contra | 3.62 | 44 | −62 | 4 | 7 | 0.59 | 0.0004 | 3.25 | 21 | −52 | 34 | 6 | 1.41 | 0.0814 |
| | Putamen | Contra | 3.68 | 22 | 6 | 6 | 8 | 0.55 | 0.0001 | 3.30 | 17 | −17 | 18 | 8 | 1.20 | 0.1656 |
| | Thalamus | Contra | 3.51 | 14 | −14 | 0 | 9 | 0.50 | 0.0010 | 3.58 | 6 | −16 | 15 | 4 | 1.91 | 0.0592 |
| | Insula | Ipsi | 4.67 | −38 | −18 | 14 | 10 | 0.49 | 0.0001 | 2.59 | −26 | −14 | 14 | 5 | 1.69 | 0.1015 |
| | Supplementary motor area | Contra | 3.91 | 8 | 4 | 46 | 11 | 0.39 | 0.0008 | 3.50 | 6 | −18 | 48 | 7 | 1.23 | 0.2315 |
| Non intensity encoding regions (in adults) | Cerebellum | Ipsi | 3.88 | −20 | −66 | −44 | | 0.35 | 0.0029 | 3.53 | −3 | −46 | −6 | | 3.57 | 0.0164 |
| | Temporal gyrus | Ipsi | 3.72 | −52 | −56 | 10 | | 0.18 | 0.5487 | 3.41 | −32 | −22 | 14 | | 2.90 | 0.0196 |
| | Supramarginal gyrus | Ipsi | 4.59 | −64 | −28 | 20 | | 0.51 | 0.0035 | 3.13 | −31 | −24 | 30 | | 2.79 | 0.0055 |
| | Cerebellum | Contra | 3.36 | 20 | −70 | −50 | | 0.31 | 0.0246 | 3.16 | 2 | −44 | −6 | | 2.72 | 0.1634 |
| | Opercular cortex | Ipsi | 5.23 | −50 | −28 | 26 | | 0.50 | 0.0018 | 2.69 | −27 | −12 | 13 | | 2.23 | 0.0710 |
| | Postcentral gyrus | Ipsi | 4.71 | −62 | −18 | 24 | | 0.44 | 0.0375 | 3.52 | −31 | −15 | 41 | | 2.12 | 0.0845 |
| | Thalamus | Ipsi | 3.52 | −12 | −14 | 10 | | 0.42 | 0.0018 | 3.48 | −1 | −20 | 13 | | 1.67 | 0.1009 |
| | Angular gyrus | Ipsi | 3.59 | −58 | −50 | 18 | | 0.53 | 0.0107 | 2.98 | −23 | −39 | 33 | | 1.56 | 0.0528 |
| | Precentral gyrus | Ipsi | 4.01 | −58 | 0 | 10 | | 0.43 | 0.0578 | 3.46 | −23 | −17 | 48 | | 1.53 | 0.1247 |
| | Frontal gyrus | Contra | 3.88 | 58 | 12 | 0 | | 0.56 | 0.0212 | 3.11 | 11 | −12 | 48 | | 1.42 | 0.0646 |
| | Cingulate gyrus | Posterior | 3.71 | −14 | −28 | 38 | | 0.08 | 0.2480 | 3.18 | −9 | −23 | 35 | | 1.42 | 0.1101 |
| | Angular gyrus | Contra | 3.71 | 60 | −46 | 18 | | 0.54 | 0.0080 | 3.12 | 22 | −51 | 35 | | 1.42 | 0.0407 |
| | Precuneous cortex | Contra | 3.60 | 16 | −68 | 40 | | 0.38 | 0.0714 | 3.70 | 5 | −30 | 52 | | 1.19 | 0.1623 |
| Active regions in both adults and infants | Visual cortex | Ipsi | 3.82 | −52 | −70 | 10 | | −0.09 | 0.3758 | 2.59 | −7 | −40 | 11 | | 1.17 | 0.1657 |
| | Brainstem | | 3.86 | 10 | −26 | −8 | | 0.33 | 0.1710 | 2.99 | −3 | −27 | −10 | | 1.11 | 0.4350 |
| | Parietal lobule | Contra | 3.10 | 20 | −44 | 68 | | 0.61 | 0.1097 | 3.10 | 27 | −24 | 46 | | 1.09 | 0.1271 |

Table 1. Continued on next page

*Table 1. Continued*

| Anatomical area | Region | Adults Peak Z within cluster | MNI coords x | y | z | Rank | Slope of regression (*E-03) | P val* | Infants Peak Z within cluster | Neonate template coords x | y | z | Rank | Slope of regression (*E-03) | P val* |
|---|---|---|---|---|---|---|---|---|---|---|---|---|---|---|---|
| Putamen | Ipsi | 3.63 | −16 | 10 | −2 | | 0.45 | 0.0023 | 3.13 | −14 | −14 | 19 | | 0.92 | 0.2813 |
| Supplementary motor area | Ipsi | 3.55 | −6 | 4 | 44 | | 0.40 | 0.0219 | 3.16 | −4 | −10 | 46 | | 0.91 | 0.3903 |
| Precentral Gyrus | Contra | 4.05 | 58 | 4 | 8 | | 0.44 | 0.0276 | 3.76 | 6 | −20 | 53 | | 0.88 | 0.2672 |
| Frontal gyrus | Ipsi | 3.57 | −8 | 22 | 32 | | −0.24 | 0.1954 | 2.79 | −13 | −9 | 50 | | 0.70 | 0.4820 |
| Pallidum | Contra | 3.40 | 16 | −4 | −4 | | 0.49 | 0.0071 | 2.84 | 13 | −13 | 13 | | 0.64 | 0.4863 |
| Amygdala | Contra | 3.49 | 20 | −2 | −14 | | 0.69 | 0.0160 | | | | | | | |
| Amygdala | Ipsi | 4.28 | −20 | −2 | −12 | | 0.43 | 0.0860 | no activity | | | | | | |
| **Active regions in adults only** Orbitofrontal cortex | Ipsi | 3.40 | −18 | 4 | −16 | | 0.42 | 0.0157 | | | | | | | |
| Orbitofrontal cortex | Contra | 3.57 | 34 | 30 | −2 | | 0.44 | 0.0460 | | | | | | | |
| Precuneous cortex | Ipsi | | | | | | | | 3.80 | −1 | −26 | 52 | | 1.26 | 0.1699 |
| Pallidum | Ipsi | | | | | | | | 3.16 | −8 | −5 | 14 | | 0.59 | 0.4787 |
| Parietal lobule | Ipsi | | | | | | | | 3.31 | −28 | −23 | 33 | | 0.99 | 0.2711 |
| Auditory cortex | Contra | no activity | | | | | | | 2.89 | 26 | −14 | 18 | | 3.07 | 0.0119 |
| Auditory cortex | Ipsi | | | | | | | | 3.34 | −17 | −29 | 19 | | 2.56 | 0.0304 |
| **Active regions in infants only** Caudate | Contra | | | | | | | | 3.61 | 13 | −17 | 22 | | 0.59 | 0.5822 |
| Caudate | Ipsi | | | | | | | | 3.47 | −7 | −8 | 18 | | 1.05 | 0.3415 |
| Hippocampus | Contra | | | | | | | | 2.61 | 21 | −25 | 9 | | 1.84 | 0.1288 |
| Hippocampus | Ipsi | | | | | | | | 2.77 | −15 | −31 | 9 | | 1.00 | 0.3326 |
| Parahippocampus | Contra | | | | | | | | 3.02 | 11 | −23 | 0 | | 1.53 | 0.3740 |
| Parahippocampus | Ipsi | | | | | | | | 2.99 | −7 | −24 | −8 | | 0.19 | 0.9013 |

Active brain regions were defined as regions with more than one active voxel with significant positive BOLD activity (z = 2.3; corrected cluster significance level of p < 0.05). The intensity encoding regions are reported with the corresponding p values and slope of the regression that refer to the degree of intensity encoding. The intensity encoding regions (in adults) are ranked according to the slope of the regression. * Threshold for significant intensity encoding was p < 0.00156 following a Bonferroni correction.

inference. The patterns of brain activity observed in this study make it likely that the infant experience is similar to that described by adults.

Pain is defined as an unpleasant sensory and emotional experience. This study provides the first demonstration that many of the brain regions that encode pain in adults are also active in full-term newborn infants within the first 7 days of life. This strongly supports the hypothesis that infants are able to experience both sensory and affective aspects of pain, and emphasizes the importance of effective clinical pain management.

## Materials and methods

### Participants

10 healthy adults (mean age = 28.3 years; range: 23–36) and 10 healthy term-born infants (mean gestational age at time of study = 40.6 weeks; range: 38.6–42.7) participated in the study. Adult participants were members of staff or postgraduate students at The University of Oxford, and infants were recruited from the Maternity Unit at the John Radcliffe Hospital, Oxford. At the time of study all infants were less than 7 days old (mean postnatal age = 3 days; range: 1–6). Infant participants were eligible for inclusion in the study if they were healthy, had no history of neurological problems, born after 37 weeks gestation, self-ventilating in air and clinically stable at the time of study.

### Recruitment

Informed written consent and consent to publish the results were provided by adult participants or by the infant's parent before the study commenced. The study was approved by the National Research Ethics Service and the University of Oxford Central University Research Ethics Committee. The study conformed to the standards set by the Declaration of Helsinki and Good Clinical Practice guidelines.

A member of the research team identified infants who were eligible for inclusion in the study shortly after birth. Prior to obtaining consent for infants to take part, parents were shown the experimental stimulators and given the opportunity to test the stimulators before they were applied to the infants. A full description of the MRI scanning environment was also provided. Parents of 113 infants were approached to take part in the study. 44 parents expressed an interest in the proposed research and 11 infants were recruited to the study. Parents were invited to stay with their infants during the study and in nearly all cases, one or both parents chose to do so. Parents were also informed that if their infant became restless while in the scanner, the study would be stopped.

Recruitment success rate was highly dependent on infant availability during the pre-booked MRI scan slots. One study was stopped due to the baby being restless when placed on the MRI bed. In adults, 100% of the subjects (10 out of 10) who were approached to take part in the study gave their consent for the psychophysical and MRI aspects of the study.

### Experimental study design

Functional magnetic resonance imaging (fMRI) of the brain was performed on all participants in response to acute noxious stimulation. On a second test occasion in the adults, the experimental protocol was repeated outside the scanner and the psychophysical data were recorded. During this session, participants were asked to verbally rate pain intensity using a numerical scale (0–10) and to describe the type of pain they experienced using the McGill pain questionnaire (*Melzack and Torgerson, 1971*).

### Experimental techniques

#### Noxious stimulation

Acute noxious (non-skin-breaking) stimulation was applied using graded nociceptive stimulators (PinPrick Stimulators, MRC Systems). In adults, five intensities of stimulation were applied to the dorsum of the left foot (applied force: 32, 64, 128, 256, and 512 mN). In infants, three intensities of stimulation were applied to the heel of the left foot (applied force: 32, 64, 128 mN). Greater force was not applied in infants to avoid the potential risk of tissue damage. Each stimulus was delivered 10 times with a minimum inter-stimulus interval (ISI) of 25 s. In all cases, stimuli were delivered by the experimenter in one smooth motion and lasted approximately 1 s.

## Recording techniques

### MRI study protocol

#### Preparation

All MRI scans conformed to the FMRIB (Functional Magnetic Resonance Imaging of the Brain) Centre ethical and safety guidelines. Adult participants were screened for MRI safety by a radiographer. Ear protection was provided (foam ear plugs, 3M, St. Paul, Minnesota; sound attenuation 28 dB) and adults were made comfortable while lying on the MRI scanner bed. The head was positioned inside the head coil and padding was used to restrict head movement.

Infants were accompanied and transported to FMRIB by a member of clinical staff, trained in neonatal life support, who remained with each infant throughout the study to ensure the infant's safety and wellbeing. Infants were screened for metal items (including metal poppers on clothing that were in direct contact with their skin) and were fed and swaddled before being placed on a vacuum-positioning mattress on the MRI bed. Ear putty, ear muffs (Minimuffs, Natus Medical Inc., Galway, Ireland), and ear-defenders (Em's 4 Bubs Baby Earmuffs, Em's 4 Kids, Brisbane, Australia) were fitted (sound attenuation levels: 23 dB, 7 dB, and 22 dB, respectively). Finally, extra padding was placed around the ear-defenders to restrict head movement. The infant's temperature was measured before the scan commenced, and heart rate and oxygen saturation was monitored throughout the scan using a 3T MRI compatible neonatal monitoring probe placed on the right foot (Fibre Optic Pulse Oximeter; Nonin Medical, Plymouth, Minnesota). Parents who accompanied their infants were also MRI safety screened and provided with adequate ear protection, and were asked to sit inside the MRI scan room throughout the scans.

#### MR image acquisition

MRI data were acquired using a Siemens 3-Tesla Magnetom Verio system (Erlangen, Germany) with a 32-channel head coil. Anatomical scans were first acquired and if excessive motion was identified, a second acquisition was attempted. For adults, a T1-weighted sequence (MPRAGE; TR = 2040 ms; TE = 4.7 ms; flip angle 8°; resolution 1 × 1 × 1 mm; axial slices = 192) was acquired and for infants a T2-weighted sequence (TSE; TR = 13871 ms; TE = 89 ms; flip angle 150°; resolution 1 × 1 × 1 mm; slices = 80) was used. BOLD images were acquired using a T2* weighted echo-planar imaging (EPI) sequence with an echo time (TE) optimised for either adults (TR = 3280 ms; TE = 30 ms; flip angle = 90°; FOV = 192 mm; imaging matrix 64 × 64; resolution 3 × 3 × 3 mm; slices = 50; average total volumes = 96) or infants (TR = 2500 ms; TE = 40 ms; flip angle = 90°; FOV = 192 mm; imaging matrix 64 × 64; resolution 3 × 3 × 3 mm; slices = 33; average total volumes = 136). Field map images were obtained for post-acquisition correction of gradient field effects. Prospective Acquisition Correction for head motion (PACE) was applied during all EPI scans. PACE is a motion correction technique that tracks the position of the head during scan acquisition and applies a real-time correction for large head movements (*Thesen et al., 2000*). The noxious stimuli were time-locked to the fMRI recording using Neurobehavioural Systems (Presentation) software that recorded each time the experimenter pressed a button while simultaneously applying the experimental stimuli to the participant's foot.

The MR data acquisition protocol was 28 min in infants and 40 min in adults. On average infants spent 60 min in the scanner room, which allowed time to prepare and settle the infants before and during scanning.

#### Sleep state

Infant sleep state could not be controlled during the study as infants fluctuated between being quietly awake and asleep. Adults were not instructed to stay awake during scanning and three adults reported that they fell asleep.

#### Adult psychophysics and pain questionnaire

Participants were asked to lie down on a patient bed. Throughout the experiment, adults were asked to verbally state a pain score following each individual stimulus using a pain scoring system where 0 is no pain and 10 is the worst pain imaginable. Once all stimuli had been presented, the participants were asked to describe the type of pain they experienced by completing the McGill Pain Questionnaire (*Melzack and Torgerson, 1971*).

## Data analysis

### MR data

All MR data processing was done using the FMRIB Software Library (FSL) (www.fmrib.ox.ac.uk/fsl). FSL Version 4.9.1 (with no boundary based registration [BBR]) was used in infants and FSL Version 5.0 was used in adults (*Woolrich et al., 2009*). Standard preprocessing steps were performed in all fMRI data sets using the FMRI Expert Analysis Tool (FEAT, version 6.0). The FSL Brain Extraction Tool (BET) was used to remove non-brain structures from the adult and infant structural images and from the adult field map images (*Smith, 2002*). In the infant field maps, brain extraction was achieved using a mask of the infant's brain-extracted structural image to guide the field map preparation. For each adult, the functional data were registered using a two-step registration: (i) the EPI image was registered to the subject's T1-weighted structural image, with a rigid body transformation, six DOF and BBR, using FMRIBs Linear Image Registration Tool (FLIRT) (*Jenkinson and Smith, 2001*; *Jenkinson et al., 2002*; *Greve and Fischl, 2009*); and (ii) the T1-weighted structural image was registered to a standard MNI image (http://www.bic.mni.mcgill.ca/ServicesAtlases/ICBM152NLin6) using FMRIBs Non-linear Registration Tool (FNIRT) with a non-linear transformation and 12 DOF. In each infant, the functional data were registered using a three step registration: (i) the EPI image was registered to the subjects T2-weighted structural images using FLIRT, with a rigid body transformation with six DOF and no BBR (*Jenkinson and Smith, 2001*; *Jenkinson et al., 2002*); (ii) the T2-weighted structural images were registered to a neonatal specific template image, which corresponded to the gestational age of the infant at the time of the study (*Serag et al., 2012*); and (iii) the template images were then registered to a standardized infant template, corresponding to a 40-week gestation infant (*Serag et al., 2012*). The final two stages of the infant registration were carried out using a non-linear transformation (FNIRT) and 12 DOF. The 40-week gestation template was chosen because it most closely matched the median age of the infants.

Functional data were spatially smoothed (full width half maximum = 5 mm) and temporal filtering (high pass cut off = 90 s) was also applied. Motion artifacts were minimised using Motion Correction with MCFLIRT (*Jenkinson et al., 2002*) and by the addition of motion-derived explanatory variables (EV) in the models. A single EV was included for each volume that was identified as having a large deviation in head position (FSL motion outliers were calculated per data set using a framewise displacement), effectively removing the signals associated with the identified timepoint from the analysis. Probabilistic independent component analysis was applied using MELODIC (model-free fMRI analysis using probabilistic independent component analysis) and components resembling movement were removed.

Time-series analysis was performed using a general linear model (GLM) by convolving the experimental design with either a standard adult hemodynamic response function (HRF) or a neonatal-specific HRF (*Arichi et al., 2012*). This approach was used to identify voxels in the brain that have a significantly increased level of BOLD activity (threshold at $z = 2.3$ with a corrected cluster significance level of $p < 0.05$). Group analyses were performed separately on adults and infants, and were performed independently for each stimulus intensity.

A voxel-based conjunction analysis was not performed between adult and infant participants because of extreme differences between infant and adult brain anatomy, which would make such an analysis unreliable. As the insula is a key region of interest in nociceptive processing the distribution of activity within the insula was also reported.

### Identifying active regions

Anatomical brain regions were classified using the Adult Harvard–Oxford cortical and subcortical atlases (*Desikan et al., 2006*) and a neonatal-specific atlas, which uses similar anatomical nomenclature as the Harvard–Oxford atlases (*Shi et al., 2011*). As the cerebellum was not identified in either the adult or neonatal atlas, and the brainstem not identified in the neonatal atlas, therefore masks (which were available as part of the standard templates in each population) were used to identify these regions. Comparisons between the infant and adult brain activity was considered on a gross anatomical scale. For example, the temporal gyrus, visual cortex, and brainstem were each considered as single structures. Active brain regions were defined as regions with more than one active voxel with significant positive BOLD activity (thresholded at $z = 2.3$ with a corrected cluster significance level of $p < 0.05$). A conversion from standard to functional space was performed to calculate the number of active voxels.

In adults, the FSL function *Atlasquery* was used to generate a list of active brain regions at each stimulus intensity, based on the masked clusters in the Harvard–Oxford cortical and subcortical atlases (*Desikan et al., 2006*). The use of *Atlasquery* ensured that all active voxels were identified in each brain region. The percentage of active voxels in each anatomical mask was then used to threshold regions, such that only regions with more than one active voxel were identified as active. The FSL function *Cluster* was used to identify the peak z statistic and MNI coordinates in each active region (see *Table 1*).

In infants, the active brain regions were identified using *MATLAB*. Infant activity at each stimulus intensity level (thresholded at z = 2.3 with a corrected cluster significance level of p < 0.05) was used to mask the neonatal atlas (*Shi et al., 2011*). The masked image was imported into *MATLAB* so that each active region could be identified and the number of active voxels within each region calculated in the neonatal atlas space. A conversion from standard to functional space allowed quantification of the number of active voxels in the infant functional space and brain regions with more than one active voxel were identified as active. The FSL function *Cluster* was used to identify peak z statistics and coordinates in neonatal template space for each active region (*Table 1*).

### Percentage BOLD increase in active anatomical brain regions

Once the active brain regions were identified at each stimulus intensity, an activity mask was created for each brain region based on the group analysis of all inputs across all stimulus intensities (z threshold = 2.3) for both the adults and infants. The activity mask was separated into anatomical regions of interest (based on brain regions which had been identified as active) and using *Featquery* the parameter estimate of the average percentage BOLD increase within each mask for each participant at each stimulus intensity was calculated.

## Statistics

### MRI data—intensity analysis

Regression analysis was carried out using the software packages Graphpad Prism & R. Mean percentage signal change was plotted against stimulus intensity and regression analysis was used to test the null hypothesis that no intensity encoding was present within the masked activity within each anatomical brain region. A Bonferroni correction for multiple comparisons was used to determine the p threshold required in order to reject the null hypothesis. The corrected significance threshold was p = 0.00156. Parameter estimates for the gradient of the regression were used to rank brain regions that were active and exhibited significant intensity encoding.

### Adult psychophysics

The mean pain score across each train of 10 stimuli at each stimulus intensity was calculated. The relationship between the mean pain scores and stimulus intensity was quantified using linear regression.

## Acknowledgements

This work was funded by the Wellcome Trust. Sezgi Goksan is a MRC funded DPhil student. We would like to thank Eugene Duff, Jelena Bozek Mouthuy, Gabriela Schmidt Mellado, Sheula Barlow, Gabrielle Green, Falk Eippert, David Parker, and Caroline Young for their analytical, clinical and technical support. We would also like to thank the infants, adults, and parents who took part in this study.

## Additional information

### Funding

| Funder | Grant reference | Author |
|---|---|---|
| Wellcome Trust | Wellcome Trust Career Development Fellowship, WT095802MA | Rebeccah Slater |
| Medical Research Council (MRC) | Graduate Student Fellowship | Sezgi Goksan |

The funders had no role in study design, data collection and interpretation, or the decision to submit the work for publication.

## Author contributions

SG, RS, Conception and design, Acquisition of data, Analysis and interpretation of data, Drafting or revising the article; CH, RP, Acquisition of data, Analysis and interpretation of data, Drafting or revising the article; FE, Conception and design, Acquisition of data; NC, FM, Analysis and interpretation of data, Drafting or revising the article; RR, EA, IT, Conception and design, Analysis and interpretation of data, Drafting or revising the article; JC, MS, Acquisition of data, Drafting or revising the article; SC, Conception and design, Acquisition of data, Analysis and interpretation of data; MJ, Acquisition of data, Analysis and interpretation of data

## Author ORCIDs

Caroline Hartley, http://orcid.org/0000-0002-7981-0836

## Ethics

Human subjects: Informed written consent and consent to publish was provided by adult participants or by the infant's parents. The study was approved by the Oxford and South Central Research Ethics Committees of the National Research Ethics Service and the University of Oxford Central University Research Ethics Committee (CUREC) (refs.: Investigating pain in the developing human brain; study number: 12/SC/0447; Human pain perception; study number: 11/SC/0249; CUREC study number: MSD/IDREC/C1/2011/143). The study conformed to the standards set by the Declaration of Helsinki and Good Clinical Practice guidelines.

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
