## [Decision Letter]

Thank you for submitting your work entitled “FMRI reveals neural activity overlap between adult and infant pain” for further consideration at *eLife*. Your manuscript was evaluated by three reaaaviewers (including Ruth Grunau and Peggy Mason, who agreed to have their identities revealed), a Reviewing editor (Jody Culham), and a Senior editor (Eve Marder).

While all involved thought the paper was interesting, timely, and important, the manuscript evoked somewhat differing views and interesting discussion amongst the reviewers through the online forum. While the reviews were largely favourable, one reviewer had strong reservations about the framing and conclusions of the paper and, relatedly, another thought the paper needed to discuss its limitations. The Reviewing editor shares these concerns. Some of these issues were brought up in the initial consultation regarding the paper, but it was thought that, as they do not require new data, they could be addressed in a revision. The full review process has reinforced the need for a substantive revision to the conclusions (and perhaps also some adjustment of the Introduction) prior to acceptance in *eLife*.

The main debate amongst the reviewing team was the degree to which one can infer pain from imaging vs. behavior. While one reviewer thought behavioral measures should be “the gold standard”, another argued that behavioral measures also have their limitations as they have only modest (∼.30) correlations with autonomic, hormonal and brain measures. No one doubted that this paper makes an important contribution in highlighting the need to revisit the question of infant pain. However, given other comments regarding the sparseness of the Discussion in the original version and the fact that *eLife* has no formatting constraints to restrict the Discussion, we agreed that the best solution would be to have you add further discussion about the limitations of inferring pain using imaging (and perhaps also behavior) in those who cannot verbally communicate.

This is one example of many caveats in using reverse inference of imaging data (e.g., Poldrack, 2006, TICS). Pain may be one of the systems where reverse inference is more successful than others ([40], Nature Methods), but even then and even with perfect data, inferences about subjective experiences remain highly speculative. Here, the absence of activation in key regions of the pain network in infants—notably the amygdala, rostral insula, and orbitofrontal cortex—mean that even greater caution needs to be applied in drawing conclusions from this data.

The following main changes are required:

1) The Discussion should be expanded to encompass the caveats of the approach (reverse inference) and present data (particularly the absence of key areas implicated in pain). The Discussion should also include additional consideration of the finding that several core areas were not activated and how that may constrain the inferential conclusion. You wish to take into account some of the specific comments appended below. You should also reconsider whether the Introduction can be adapted to reduce the potential perception that it is ”polemical”.

2) To provide a measure of the specificity, and rule out any global effects of the pain stimulus in infants, it would be helpful to include some control regions that do not show a pain effect in adults, and report their (lack of) response in infants using the same methods as for the pain regions.

Further details:

Although the *eLife* policy is to summarize the main points of the reviews, in some cases, including this one, the Reviewing editor likes to make the full commentary of substantive points available to the authors. As such, here are specific points and discussion that the authors should consider in writing the revision:

*Reviewer #2*:

This is an interesting study on an important topic. The issue of how neonates experience noxious stimulation is of clear significance with major policy implications. Thus this study is extremely well motivated.

Unfortunately, the authors appear to have come into the study with a preferred answer. The Introduction reads as polemical and the interpretations are not the most straightforward ones given the experimental findings.

The Introduction is inappropriately polemical, strongly arguing that infants feel pain as do adults. Yet this issue is the question addressed by the experiments reported in this paper. If it is so clear that infants feel pain in an adult fashion and that under-medication is therefore inhumane, then why do this study?

Behavior is the gold standard over brain activation. However, the authors reverse this, arguing that as long as there is brain activation in adult regions, then the experience is similar and thus infants experience the pain percept as do adults. Further the authors hold that this is true even though the infants do not show behavioral signs of escape or distress. As Stuart Derbyshire stresses, infants need to learn to associate or contextualize the aversive experience with the somatosensory activation. This is a learned association. The results suggest that the babies tested have not yet made such an association.

Infants withdrew at all intensities and adults at none. It should be noted that Jens Schoenberg has clearly demonstrated that low threshold tactile inputs elicit withdrawals in neonates (rodents). Schoenberg demonstrates that this arrangement allows the somatosensory system to develop correctly with the appropriate input-output connections. It is unclear to this reviewer how the authors interpret this finding.

No activity in amygdala, rostral insula, or orbitofrontal cortex. The authors correctly interpret this: “It is likely that the infants are too immature and inexperienced to evaluate and contextualise the nociceptive stimulus into a coordinated decision and response, which might account for the lack of activity within these regions. Similarly, in adults the amygdala is thought to attach emotional significance to the nociceptive inputs it receives, and to play a role in fear and anxiety (Simons, 2014), which may reflect affective qualities that the newborn infant does not yet ascribe to the stimulus.” Bingo. Given this (I believe correct) interpretation, then the conclusion should be the opposite of what the authors make it out to be. In short, the neonates do not appear to experience the aversive qualities of noxious stimulation.

*Reviewer #1 comments resulting from our discussion*:

Behavior is not a gold standard for pain. This was the view historically, but is no longer the case. Currently it is recognized that there is no gold standard for infant pain.

In infants:

1) Correlations between behavioural, autonomic, hormonal and brain parameters are modest, around .30. These various parameters provide complementary information about pain and stress in infants who cannot communicate verbally;

2) Newborn infant pain response is variable, depending on multiple factors including sleep/wake state, prior events and context. In the MRI scanner, for the least movement artifacts, it is optimal if the infant is asleep. However, it has been established that fewer facial behavioral responses are seen during procedural pain in sleeping infants.

---

## [Author Response]

We have addressed the two keys issues raised in your editorial letter, which primarily required a more detailed discussion of the data. In brief, we have discussed the limitations of using a ‘reverse inference’ approach to interpret brain imaging findings; have discussed how the lack of infant amygdala, OFC and anterior insula activation could alter the subjective experience of pain; and have included examples of brain regions that were not activated by the noxious stimulation in infants. We have also removed some (relatively outdated) references from the Introduction, which may have given the impression that the Introduction was biased.

In addition, we have also addressed the minor comments raised by the reviewers. We have highlighted the benefits of using a multi-dimensional approach to the assessment of infant pain; emphasised that these studies were conducted in newborn infants in the first seven days of life; and described the lack of certainty regarding whether observed differences between infant and adult activation patterns are due to differences in peripheral drive or subsequent central processing.